# Evaluating State-of-the-Art Classification Models Against Bayes Optimality

**Ryan Theisen**[*][†]
University of California, Berkeley
theisen@berkeley.edu

**Huan Wang**[*]
Salesforce Research
huan.wang@salesforce.com

**Lav R. Varshney**[†]
University of Illinois Urbana-Champaign
varshney@illinois.edu

**Caiming Xiong**
Salesforce Research
cxiong@salesforce.com

**Richard Socher**[†]
you.com
rsocher@gmail.com

## Abstract

Evaluating the inherent difficulty of a given data-driven classification problem is important for establishing absolute benchmarks and evaluating progress in the field. To this end, a natural quantity to consider is the *Bayes error*, which measures the optimal classification error theoretically achievable for a given data distribution. While generally an intractable quantity, we show that we can compute the exact Bayes error of generative models learned using normalizing flows. Our technique relies on a fundamental result, which states that the Bayes error is invariant under invertible transformation. Therefore, we can compute the exact Bayes error of the learned flow models by computing it for Gaussian base distributions, which can be done efficiently using Holmes-Diaconis-Ross integration. Moreover, we show that by varying the temperature of the learned flow models, we can generate synthetic datasets that closely resemble standard benchmark datasets, but with almost any desired Bayes error. We use our approach to conduct a thorough investigation of state-of-the-art classification models, and find that in some — but not all — cases, these models are capable of obtaining accuracy very near optimal. Finally, we use our method to evaluate the intrinsic "hardness" of standard benchmark datasets.

## 1 Introduction

Benchmark datasets and leaderboards are prevalent in machine learning's common task framework [8]; however, this approach inherently relies on relative measures of improvement. It may therefore be insightful to be able to evaluate state-of-the-art (SOTA) performance against the optimal performance theoretically achievable by *any* model [34]. For supervised classification tasks, this optimal performance is captured by the Bayes error rate which, were it tractable, would not only give absolute benchmarks, rather than just comparing to previous classifiers, but also insights into dataset hardness [14, 38] and which gaps between SOTA and optimal the community may fruitfully try to close.

Suppose we have data generated as $(X, Y) \sim p$, where $X \in \mathbb{R}^d$, $Y \in \mathcal{Y} = \{1, \dots, K\}$ is a label and $p$ is a distribution over $\mathbb{R}^d \times \mathcal{Y}$. The **Bayes classifier** is the rule which assigns a label to an

---

[*]Equal contribution.
[†]Work done while at Salesforce Research.

35th Conference on Neural Information Processing Systems (NeurIPS 2021).

observation $\mathbf{x}$ via

$$y = C_{\text{Bayes}}(\mathbf{x}) := \arg\max_{j \in \mathcal{Y}} p(Y = j \mid X = \mathbf{x}). \tag{1}$$

The **Bayes error** is simply the probability that the Bayes classifier predicts incorrectly:

$$\mathcal{E}_{\text{Bayes}}(p) := p(C_{\text{Bayes}}(X) \neq Y). \tag{2}$$

The Bayes classifier is optimal, in the sense it minimizes $p(C(X) \neq Y)$ over all possible classifiers $C : \mathbb{R}^d \to \mathcal{Y}$. Therefore, the Bayes error is a natural measure of 'hardness' of a particular learning task. Knowing $\mathcal{E}_{\text{Bayes}}$ should interest practitioners: it gives a natural benchmark for the performance of any trained classifier. In particular, in the era of deep learning, where vast amounts of resources are expended to develop improved models and architectures, it is of great interest to know whether it is even theoretically possible to substantially lower the test errors of state-of-the-art models, cf. [6].

Of course, obtaining the exact Bayes error will almost always be intractable for real-world classification tasks, as it requires full knowledge of the distribution $p$. A variety of works have developed estimators for the Bayes error, either based on upper and/or lower bounds [2] or exploiting exact representations of the Bayes error [26, 24]. Most of these bounds and/or representations are in terms of some type of *distance* or *divergence* between the class conditional distributions,

$$p_j(\mathbf{x}) := p(X = \mathbf{x} \mid Y = j), \tag{3}$$

and/or the marginal label distributions $\pi_j := p(Y = j)$. For example, there are exact representations of the Bayes error in terms of a particular $f$-divergence [26], and in a special case in terms of the total variation distance [24]. More generally, there are lower and upper bounds known for the Bayes error in terms of the Bhattacharyya distance [2, 24], various $f$-divergences [20], the Henze-Penrose (HP) divergence [22, 21], as well as others. Once one has chosen a desired representation and/or bound in terms of some divergence, estimating the Bayes error reduces to the estimation of this divergence. Unfortunately, for high-dimensional datasets, this estimation is highly inefficient. For example, most estimators of $f$-divergences rely on some type of $\varepsilon$-ball approach, which requires a number of samples on the order of $(1/\varepsilon)^d$ in $d$ dimensions [26, 30]. In particular, for large benchmark image datasets used in deep learning, this approach is inadequate to obtain meaningful results.

Here, we take a different approach: rather than computing an approximate Bayes error of the exact distribution (which, as we argue above, is intractable in high dimensions), we propose to compute the *exact Bayes error of an approximate distribution*. The basics of our approach are as follows.

- We show that when the class-conditional distributions are Gaussian $q_j(\mathbf{z}) = \mathcal{N}(\mathbf{z}; \boldsymbol{\mu}_j, \boldsymbol{\Sigma})$, we can efficiently compute the Bayes error using a variant of Holmes-Diaconis-Ross integration proposed in [12].

- We use normalizing flows [28, 16, 9] to fit approximate distributions $\hat{p}_j(\mathbf{x})$, by representing the original features as $\mathbf{x} = T(\mathbf{z})$ for a learned invertible transformation $T$, where $\mathbf{z} \sim q_j(\mathbf{z}) = \mathcal{N}(\mathbf{z}; \boldsymbol{\mu}_j, \boldsymbol{\Sigma})$, for learned parameters $\boldsymbol{\mu}_j, \boldsymbol{\Sigma}$.

- Lastly, we prove in Proposition 1 that the Bayes error is invariant under invertible transformation of the features, so computing the Bayes error of the approximants $\hat{p}_j(\mathbf{x})$ can be done *exactly* by computing it for the Gaussians $q_j(\mathbf{z})$.

Moreover, we show that by varying the *temperature* of a single flow model, we can obtain an entire class of distributions with varying Bayes errors. This recipe allows us to compute the Bayes error of a large variety of distributions, which we use to conduct a thorough empirical investigation of a benchmark datasets and SOTA models, producing a library of trained flow models in the process. By generating synthetic versions of standard benchmark datasets with known Bayes errors, and training them on SOTA deep learning architectures, we are able to assess how well these models perform compared to the Bayes error, and find that in some cases they indeed achieve errors very near optimal. We then investigate our Bayes error estimates as a measure of objective difficulty of benchmark classification tasks, and produce a ranking of these datasets based on their approximate Bayes errors.

We should note one additional point before proceeding. In general the hardness of classification tasks can be decomposed into two relatively independent components: i) hardness caused by the lack of

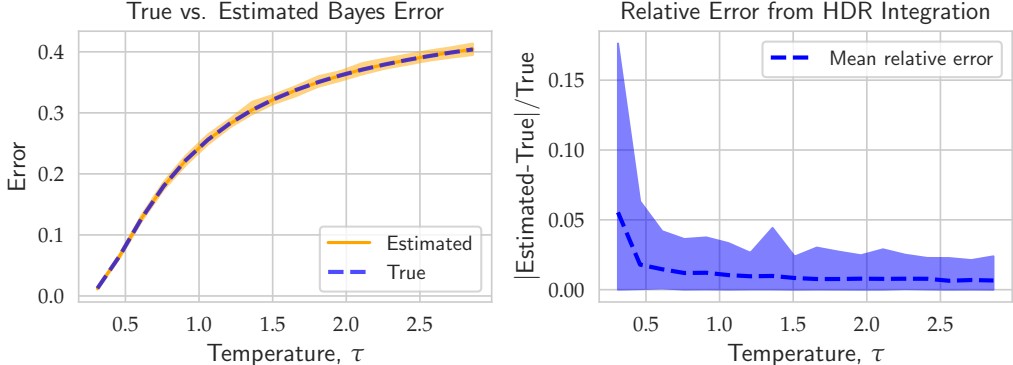

Figure 1: We compare the Bayes error estimated using HDR integration [12] with the exact error in the binary classification with equal covariance case given in (4). On the right we show the relative error from numerical integration. Shaded region on both plots shows the range over 100 runs. We see the integration routine gives highly accurate estimates. Here we use dimension $d = 784$, and take $\boldsymbol{\mu}_1, \boldsymbol{\mu}_2$ to be randomly drawn unit vectors, and $\boldsymbol{\Sigma} = \tau^2 \mathbf{I}$ where $\tau$ is the temperature.

samples, and ii) hardness caused by the internal data distribution $p$. The focus of this work is about the latter: the hardness caused by $p$. Indeed, even if the Bayes error of a particular task is known to be a particular value $\mathcal{E}_{\text{Bayes}}$, it may be highly unlikely that this error is achievable given a model trained on only $N$ samples from $p$. The problem of finding the minimal error achievable from a given dataset of size $N$ has been called the optimal experimental design problem [31]. While this is not the focus of the present work, an interesting direction for future work is to use our methodology to investigate the relationship between $N$ and the SOTA-Bayes error gap.

## 2  Computing the Bayes error of Gaussian conditional distributions

Throughout this section, we assume the class conditional distributions are Gaussian: $q_j(\mathbf{x}) = \mathcal{N}(\mathbf{z}; \boldsymbol{\mu}_j, \boldsymbol{\Sigma}_j)$. In the simplest case of binary classification with $K = 2$ classes, equal covariance $\boldsymbol{\Sigma}_1 = \boldsymbol{\Sigma}_2 = \boldsymbol{\Sigma}$, and equal marginals $\pi_1 = \pi_2 = \frac{1}{2}$, the Bayes error can be computed analytically in terms of the CDF of the standard Gaussian distribution, $\Phi(\cdot)$, as:

$$\mathcal{E}_{\text{Bayes}} = 1 - \Phi\left(\tfrac{1}{2}\|\boldsymbol{\Sigma}^{-1/2}(\boldsymbol{\mu}_1 - \boldsymbol{\mu}_2)\|_2\right). \tag{4}$$

When $K > 2$ and/or the covariances are different between classes, there is no closed-form expression for the Bayes error. Instead, we work from the following representation:

$$\mathcal{E}_{\text{Bayes}} = 1 - \sum_{k=1}^{K} \pi_k \int \prod_{j \neq k} \mathbb{1}(q_j(\mathbf{z}) < q_k(\mathbf{z})) \mathcal{N}(d\mathbf{z}; \boldsymbol{\mu}_k, \boldsymbol{\Sigma}_k). \tag{5}$$

In the general case, the constraints $q_j(\mathbf{z}) < q_k(\mathbf{z})$ are quadratic, with $q_j(\mathbf{z}) < q_k(\mathbf{z})$ occurring if and only if:

$$-(\mathbf{z} - \boldsymbol{\mu}_j)^\top \boldsymbol{\Sigma}_j^{-1}(\mathbf{z} - \boldsymbol{\mu}_j) - \log \det \boldsymbol{\Sigma}_j < -(\mathbf{z} - \boldsymbol{\mu}_k)^\top \boldsymbol{\Sigma}_k^{-1}(\mathbf{z} - \boldsymbol{\mu}_k) - \log \det \boldsymbol{\Sigma}_k. \tag{6}$$

As far as we know, there is no efficient numerical integration scheme for computing Gaussian integrals under general quadratic constraints of this form. However, if we further assume the covariances are equal, $\boldsymbol{\Sigma}_j = \boldsymbol{\Sigma}$ for all $j = 1, \ldots, K$, then the constraint (6) becomes linear, of the form

$$\mathbf{a}_{jk}^\top \mathbf{z} + b_{jk} > 0, \tag{7}$$

where $\mathbf{a}_{jk} := 2\boldsymbol{\Sigma}^{-1}(\boldsymbol{\mu}_j - \boldsymbol{\mu}_k)$ and $b_{jk} := \boldsymbol{\mu}_k^\top \boldsymbol{\Sigma}^{-1} \boldsymbol{\mu}_k - \boldsymbol{\mu}_j^\top \boldsymbol{\Sigma}^{-1} \boldsymbol{\mu}_j$. Thus expression (5) can be written as

$$\mathcal{E}_{\text{Bayes}} = 1 - \sum_{k=1}^{K} \pi_k \int \prod_{j \neq k} \mathbb{1}(\mathbf{a}_{jk}^\top \mathbf{z} + b_{jk} > 0) \mathcal{N}(d\mathbf{z}; \boldsymbol{\mu}_k, \boldsymbol{\Sigma}). \tag{8}$$

Computing integrals of this form is precisely the topic of the recent paper [12], which exploited the particular form of the linear constraints and the Gaussian distribution to develop an efficient integration scheme using a variant of the Holmes-Diaconis-Ross method [7]. This method is highly efficient, even in high dimensions[3]. In Figure 1, we show the estimated Bayes error using this method on a synthetic binary classification problem in $d = 784$ dimensions, where we can use closed-form expression (4) to measure the accuracy of the integration. As we can see, it is highly accurate.

This method immediately allows us to investigate the behavior of large neural network models on high-dimensional synthetic datasets with class conditional distributions $q_j(\mathbf{z}) = \mathcal{N}(\mathbf{z}; \boldsymbol{\mu}_j, \boldsymbol{\Sigma})$. However, in the next section, we will see that we can use normalizing flows to estimate the Bayes error of real-world datasets as well.

## 3 Normalizing flows and invariance of the Bayes error

Normalizing flows are a powerful technique for modeling high-dimensional distributions [28]. The main idea is to represent the random variable $\mathbf{x}$ as a transformation $T_\phi$ (parameterized by $\phi$) of a vector $\mathbf{z}$ sampled from some, usually simple, base distribution $q(\mathbf{z}; \psi)$ (parameterized by $\psi$), i.e.

$$\mathbf{x} = T_\phi(\mathbf{z}) \qquad \text{where} \qquad \mathbf{z} \sim q(\mathbf{z}; \psi). \tag{9}$$

When the transformation $T_\phi$ is invertible, we can obtain the exact likelihood of $\mathbf{x}$ using a standard change of variable formula:

$$\hat{p}(\mathbf{x}; \theta) = q(T_\phi^{-1}(\mathbf{x}); \psi) \left| \det J_{T_\phi}(T_\phi^{-1}(\mathbf{x})) \right|^{-1}, \tag{10}$$

where $\theta = (\phi, \psi)$ and $J_{T_\phi}$ is the Jacobian of the transformation $T_\phi$. The parameters $\theta$ can be optimized, for example, using the KL divergence:

$$\mathcal{L}(\theta) = D_{\mathrm{KL}}(p(\mathbf{x}) \parallel \hat{p}(\mathbf{x}; \theta)) \approx -\frac{1}{N} \sum_{i=1}^{N} \log q(T_\phi^{-1}(\mathbf{x}_i), \psi) + \log \left| \det J_{T_\phi^{-1}}(\mathbf{x}_i) \right| + \text{const.} \tag{11}$$

This approach is easily extended to the case of learning class-conditional distributions by parameterizing multiple base distributions $q_j(\mathbf{z}; \psi_j)$ and computing

$$\hat{p}_j(\mathbf{x}; \theta) = q_j(T_\phi^{-1}(\mathbf{x}); \psi_j) \left| \det J_{T_\phi}(T_\phi^{-1}(\mathbf{x})) \right|^{-1}. \tag{12}$$

For example, we can take $q_j(\mathbf{z}; \boldsymbol{\mu}_j, \boldsymbol{\Sigma}) = \mathcal{N}(\mathbf{z}; \boldsymbol{\mu}_j, \boldsymbol{\Sigma})$, where we fit the parameters $\boldsymbol{\mu}_j, \boldsymbol{\Sigma}$ during training. This is commonly done to learn class-conditional distributions, e.g. [16]. This is the approach we take in the present work. In practice, the invertible transformation $T_\phi$ is parameterized as a neural network, though special care must be taken to ensure the neural network is invertible and has a tractable Jacobian determinant. Here, we use the Glow architecture [16] throughout our experiments, as detailed in Section 4.

### 3.1 Invariance of the Bayes Error

Normalizing flow models are particularly convenient for our purposes, since we can prove the Bayes error is invariant under invertible transformation. This is formalized as follows.

**Proposition 1.** *Let $(X, Y) \sim p$, $X \in \mathbb{R}^d, Y \in \mathcal{Y} = \{1, \ldots, K\}$, and let $\mathcal{E}_{Bayes}(p)$ be the associated Bayes error of this distribution. Let $T : \mathbb{R}^d \to \mathbb{R}^d$ be an invertible map and denote $q$ the associated joint distribution of $Z = T(X)$ and $Y$. Then $\mathcal{E}_{Bayes}(p) = \mathcal{E}_{Bayes}(q)$.*

*Proof.* For convenience, denote $|\mathbf{A}|$ as the absolute value determinant of a matrix $\mathbf{A}$. Using the representation derived in [26], we can write the Bayes error as

$$\mathcal{E}_{\mathrm{Bayes}}(p) = 1 - \pi_1 - \sum_{k=2}^{K} \int \max \left( 0, \pi_k - \max_{1 \le i \le k-1} \pi_i \frac{p_i(\mathbf{x})}{p_k(\mathbf{x})} \right) p_k(\mathbf{x}) d\mathbf{x}. \tag{13}$$

---

[3]Note that the integrals appearing in (8) are really only $(K-1)$-dimensional integrals, since they only depend on $K-1$ variables of the form $\mathbf{a}_{jk}^\top \mathbf{x} + b_{jk}$.

Then if $\mathbf{z} = T(\mathbf{x})$, we have that $q_k(\mathbf{z}) = p_k(T(\mathbf{z}))|J_T(\mathbf{z})|$, and $d\mathbf{x} = |J_{T^{-1}}(\mathbf{z})|d\mathbf{z}$. Hence

$$\mathcal{E}_{\text{Bayes}}(p) = 1 - \pi_1 - \sum_{k=2}^{K} \int \max\left(0, \pi_k - \max_{1 \leq i \leq k-1} \pi_i \frac{p_i(\mathbf{x})}{p_k(\mathbf{x})}\right) p_k(\mathbf{x}) d\mathbf{x}$$

$$= 1 - \pi_1 - \sum_{k=2}^{K} \int \max\left(0, \pi_k - \max_{1 \leq i \leq k-1} \pi_i \frac{q_i(\mathbf{z})|J_T(\mathbf{z})|}{q_k(\mathbf{z})|J_T(\mathbf{z})|}\right) q_k(\mathbf{z})|J_T(\mathbf{z})||J_{T^{-1}}|(\mathbf{z})d\mathbf{z}.$$

By the Inverse Function Theorem, $|J_{T^{-1}}(\mathbf{z})| = |J_T(\mathbf{z})|^{-1}$, and so we get

$$\mathcal{E}_{\text{Bayes}}(p) = 1 - \pi_1 - \sum_{k=2}^{K} \int \max\left(0, \pi_k - \max_{1 \leq i \leq k-1} \pi_i \frac{q_i(\mathbf{z})|J_T(\mathbf{z})|}{q_k(\mathbf{z})|J_T(\mathbf{z})|}\right) q_k(\mathbf{z})|J_T(\mathbf{z})||J_T(\mathbf{z})|^{-1}d\mathbf{z}$$

$$= 1 - \pi_1 - \sum_{k=2}^{K} \int \max\left(0, \pi_k - \max_{1 \leq i \leq k-1} \pi_i \frac{q_i(\mathbf{z})}{q_k(\mathbf{z})}\right) q_k(\mathbf{z})d\mathbf{z}$$

$$= \mathcal{E}_{\text{Bayes}}(q),$$

which completes the proof. $\qquad\square$

This result means that we can compute the *exact* Bayes error of the approximate distributions $\hat{p}_j(\mathbf{x}; \theta)$ using the methods introduced in Section 2 with the Gaussian conditionals $q_j(\mathbf{z}; \boldsymbol{\mu}_j, \boldsymbol{\Sigma})$. If in addition the flow model $\hat{p}_j(\mathbf{x}; \theta)$ is a good a approximation for the true class-conditional distribution $p_j(\mathbf{x})$, then we expect to obtain a good estimate for the true Bayes error. In what follows, we will see examples both of when this is and is not the case.

### 3.2 Varying the Bayes error using temperature

An important aspect of the normalizing flow approach is that we can in fact generate a whole family of distributions from a single flow model. To do this, we can vary the *temperature* $\tau$ of the model by multiplying the covariance $\boldsymbol{\Sigma}$ of the base distribution by $\tau^2$ to get $q_{j,\tau}(\mathbf{z}) := \mathcal{N}(\mathbf{z}; \boldsymbol{\mu}_j, \tau^2\boldsymbol{\Sigma})$. The same invertible map $T_\phi$ induces new conditional distributions,

$$\hat{p}_{j,\tau}(\mathbf{x}; \theta) = q_{j,\tau}(T_\phi^{-1}(\mathbf{x}); \psi_j) \left|\det J_{T_\phi}(T_\theta^{-1}(\mathbf{x}))\right|^{-1}, \tag{14}$$

as well as the associated joint distribution $\hat{p}_\tau(y = j, \mathbf{x}; \theta) = \pi_j \hat{p}_{j,\tau}(\mathbf{x}; \theta)$.

It can easily be seen that the Bayes error of $\hat{p}_\tau$ is increasing in $\tau$.

**Proposition 2.** *The Bayes error of flow models is monotonically increasing in $\tau$. That is, for $0 < \tau \leq \tau'$, we have that $\mathcal{E}_{Bayes}(\hat{p}_\tau) \leq \mathcal{E}_{Bayes}(\hat{p}_{\tau'})$.*

This fact means that we can easily generate datasets of varying difficulty by changing the temperature $\tau$. For example, in Figure 2 we show samples generated by a flow model (see Section 4 for implementation details) trained on the Fashion-MNIST dataset at various values of temperature and the associated Bayes error. As $\tau \to 0^+$, the distribution $\hat{p}_{j,\tau}$ concentrate on the mode of the distributions $\hat{p}_j$, making the classification tasks easy, whereas when $\tau$ gets large, the distributions $\hat{p}_{j,\tau}$ become more uniform, making classification more challenging. In practice, this can be used to generate datasets with almost arbitrary Bayes error: for any prescribed error $\varepsilon$ in the range of the map $\tau \mapsto \mathcal{E}_{\text{Bayes}}(\hat{p}_\tau)$, we can numerically invert this map to find $\tau$ for which $\mathcal{E}_{\text{Bayes}}(\hat{p}_\tau) = \varepsilon$.

## 4 Empirical investigation

### 4.1 Setup

**Datasets and data preparation.** We train flow models[4] on a wide variety of standard benchmark datasets: MNIST [19], Extended MNIST (EMNIST) [5], Fashion MNIST [36], CIFAR-10 [17], CIFAR-100 [17], SVHN [23], and Kuzushiji-MNIST [4]. The EMNIST dataset has several different splits, which include splits by digits, letters, merge, class, and balanced. The images in MNIST, Fashion-MNIST, EMNIST, and Kuzushiji-MNIST are padded to 32-by-32 pixels.[5]

---

[4]Code can be found at https://github.com/salesforce/DataHardness.

[5]Glow implementation requires the input dimension to be power of 2.

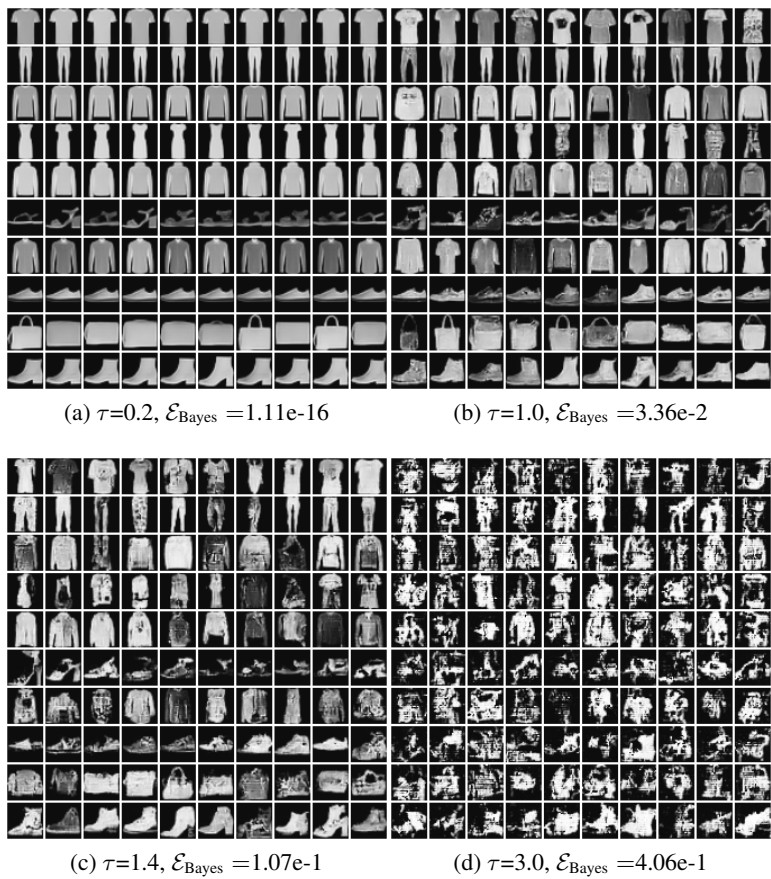

(a) $\tau$=0.2, $\mathcal{E}_{\text{Bayes}}$ =1.11e-16    (b) $\tau$=1.0, $\mathcal{E}_{\text{Bayes}}$ =3.36e-2

(c) $\tau$=1.4, $\mathcal{E}_{\text{Bayes}}$ =1.07e-1    (d) $\tau$=3.0, $\mathcal{E}_{\text{Bayes}}$ =4.06e-1

Figure 2: Generated Fashion-MNIST Samples with Different Temperatures

We remark that we observe our Bayes error estimator runs efficiently when the input is of dimension 32-by-32-by-3. However it is in general highly memory intensive to run the HDR integration routine on significantly larger datasets, e.g. when the input size grows to 64-by-64-by-3. As a consequence, in our experiments we only work on datasets of dimension no larger than 32-by-32-by-3.

**Modeling and training.** The normalizing flow model we use in our experiments is a pytorch implementation [13] of Glow [16]. In all our the experiments, affine coupling layers are used, the number of steps of the flow in each level $K = 16$, the number of levels $L = 3$, and number of channels in hidden layers $C = 512$.

During training, we minimize the Negative Log Likelihood Loss (NLL)

$$\text{NLL}(\{\mathbf{x}_i, y_i\}) = -\frac{1}{N} \sum_{i=1}^{N} \left( \log p_{y_i}(\mathbf{x}_i; \theta) + \log \pi_{y_i} \right). \tag{15}$$

As suggested in [16], we also add a classification loss to predict the class labels from the second-to-last layer of the encoder with a weight of $\lambda$. During the experiments we traversed configurations with $\lambda = \{0.01, 0.1, 1.0, 10\}$, and report the numbers produced by the model with the smallest NLL loss on the test set. Note here even though we add the classification loss in the objective as a regularizer, the model is selected based on the smallest NLL loss in the test set instead of the classification loss or the total loss. The training and evaluation are done on a workstation with 2 NVIDIA V100 GPUs.

### 4.2 Evaluating SOTA models against generated datasets

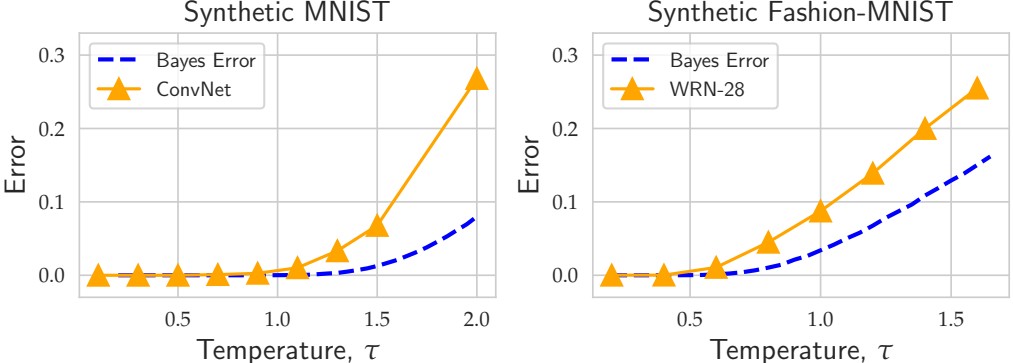

Figure 3: Test errors of synthetic versions of MNIST and Fashion-MNIST, generated at various temperatures, and their corresponding Bayes error. Here we used 60,000 training samples, and 10,000 testing samples, to mimic the original datasets. The model used in Fashion-MNIST was a Wide-ResNet-28-10, which attains nearly start of the accuracy on the original Fashion-MNIST dataset [39]. The model used in MNIST is a popular ConvNet [1].

In this section, we use our trained flow models to generate synthetic versions of standard benchmark datasets, for which the Bayes error is known exactly. In particular, we generate synthetic versions of the MNIST and Fashion-MNIST datasets at varying temperatures. As we saw in Section 3.2, varying the temperature allows us to generate datasets with different difficulty. Here, we train a Wide-ResNet-28-10 model (i.e. a ResNet with depth 28 and width multiple 10) [37, 35] on these datasets, and compare the test error to the exact Bayes error for these problems. This Wide-ResNet model (together with appropriate data augmentation) attains nearly state-of-the-art accuracy on the original Fashion-MNIST dataset [39], and so we expect that our results here reflect roughly the best accuracy presently attainable on these synthetic datasets as well. To make the comparison fair, we use a training set size of 60,000 to mimic the size of the original MNIST series of datasets.

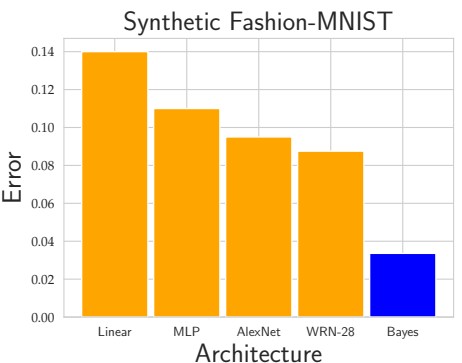

Figure 4: Errors of various model architectures (from old to modern) on a Synthetic Fashion-MNIST dataset ($\tau = 1$). We can see that for this task, while accuracy has improved with modern models, there is still a substantial gap between the SOTA and Bayes optimal.

The Bayes errors as well as the test errors achieved by the Wide-ResNet or ConvNet models are shown in Figure 3. As one would expect, the errors of trained models increase with temperature. It can be observed that Wide-ResNet and ConvNet are able to achieve close-to-optimal performance when the dataset is relatively easy, e.g., $\tau < 1$ for MNIST and $\tau < 0.5$ for Fashion-MNIST. The gap becomes more significant when the dataset is harder, e.g. $\tau > 1.5$ for MNIST and $\tau > 1$ for Fashion-MNIST.

For the Synthetic Fashion-MNIST dataset at temperature $\tau = 1$, in addition to the Wide-ResNet (WRN-28) considered above, we also trained three other architectures: a simple linear classifier (Linear), a 1-hidden layer ReLU network (MLP) with 500 hidden units, and a standard AlexNet convolutional architecture [18]. The resulting test errors, as well as the Bayes error, are shown in Figure 4. We see that while the development of modern architectures has led to substantial improvement in the test error, there is still a reasonably large gap between the performance of the SOTA Wide-ResNet and Bayes optimality. Nonetheless, it is valuable to know that, for this task, the state-of-the-art has substantial room to be improved.

| Corpus | #classes | #samples | NLL | Bayes Error | SOTA Error [29] |
|---|---|---|---|---|---|
| MNIST | 10 | 60,000 | 8.00e2 | 1.07e-4 | 1.6e-3 [3] |
| EMNIST (digits) | 10 | 280,000 | 8.61e2 | 1.21e-3 | 5.7e-3 [27] |
| SVHN | 10 | 73,257 | 4.65e3 | 7.58e-3 | 9.9e-3 [3] |
| Kuzushiji-MNIST | 10 | 60,000 | 1.37e3 | 8.03e-3 | 6.6e-3 [11] |
| CIFAR-10 | 10 | 50,000 | 7.43e3 | 2.46e-2 | 3e-3 [10] |
| Fashion-MNIST | 10 | 60,000 | 1.75e3 | 3.36e-2 | 3.09e-2 [32] |
| EMNIST (letters) | 26 | 145,600 | 9.15e2 | 4.37e-2 | 4.12e-2 [15] |
| CIFAR-100 | 100 | 50,000 | 7.48e3 | 4.59e-2 | 3.92e-2 [10] |
| EMNIST (balanced) | 47 | 131,600 | 9.45e2 | 9.47e-2 | 8.95e-2 [15] |
| EMNIST (bymerge) | 47 | 814,255 | 8.53e2 | 1.00e-1 | 1.90e-1 [5] |
| EMNIST (byclass) | 62 | 814,255 | 8.76e2 | 1.64e-1 | 2.40e-1 [5] |

Table 1: We evaluate the estimated Bayes error on image data sets and rank them by relative difficulty. Comparisons with prediction performance of state-of-the-art neural network models shows that our estimation is highly aligned with empirically observed performance.

## 4.3 Dataset Hardness Evaluation

A important application of our Bayes error estimator is to estimate the inherent *hardness* of a given dataset, regardless of model. We run our estimator on several popular image classification corpora and rank them based on our estimated Bayes error. The results are shown in Table 1. As a comparison we also put the SOTA numbers in the table.

Before proceeding, we make two remarks. First, all of the Bayes errors reported here were computed using temperature $\tau = 1$. This is for two main reasons: 1) setting $\tau = 1$ reflects the flow model attaining the lowest testing NLL, and hence is in some sense the "best" approximation for the true distribution, 2) in our experiments, the ordering of the hardness of classes is unchanged by varying temperature, and so taking $\tau = 1$ is a reasonable default. Second, the reliability of the Bayes errors reported here as a measure of inherent difficulty are dependent on the quality of the approximate distribution $\hat{p}$; if this distribution is not an adequate estimate of the true distribution $p$, then it is possible that the Bayes errors do not accurately reflect the true difficulty of the original dataset. Therefore, we also report the test NLL for each model as a metric to evaluate the quality of the approximant $\hat{p}$.

First, we observe that, by and large, the estimated Bayes errors align well with SOTA. In particular, if we constrain the NLL loss to be smaller than 1000, then ranking by our estimated Bayes error aligns exactly with SOTA.

Second, the NLL loss in MNIST, Fashion MNIST, EMNIST and Kuzushiji-MNIST is relatively low, suggesting a good approximation by normalizing flow. However corpora such as CIFAR-10, CIFAR-100, and SVHN may suffer from a lack of training samples. In general large NLL loss may be due to either insufficient model capacity or lack of samples. In our experiments, we always observe the Glow model is able to attain essentially zero error on the training corpus, so it is highly possible the large NLL loss is caused by the lack of training samples.

Third, for datasets such as MNIST, EMNIST (digits, letters, balanced), SVHN, Fashion-MNIST, Kuzushiji-MNIST, CIFAR-10, and CIFAR-100 the SOTA numbers are roughly the same order of magnitude as the Bayes error. On the other hand, for EMNIST (bymerge and byclass) there is still substantial gap between the SOTA and estimated Bayes errors. This is consistent with the fact that there is little published literature about these two datasets; as a result models for them are not as well-developed.

## 5 Limitations, Societal Impact, and Conclusion

In this work, we have proposed a new approach to benchmarking state-of-the-art models. Rather than comparing trained models to each other, our approach leverages normalizing flows and a key invariance result to be able to generate benchmark datasets closely mimicking standard benchmark datasets, but with *exactly controlled* Bayes error. This allows us to evaluate the performance of trained models on an absolute, rather than relative, scale. In addition, our approach naturally gives us

a method to assess the relative hardness of classification tasks, by comparing their estimated Bayes errors.

While our work has led to several interesting insights, there are also several limitations at present that may be a fruitful source of future research. For one, it is possible that the Glow models we employ here could be replaced with higher quality flow models, which would perhaps lead to better benchmarks and better estimates of the hardness of classification tasks. To this end, it is possible that the well-documented label noise in standard datasets contributes to our inability to learn higher-quality flow models [25]. To the best of our knowledge, there has not been significant work using normalizing flows to accurately estimate class-conditional distributions for NLP datasets; this in itself would be an interesting direction for work. Second, a major limitation of our approach is that there isn't an immediately obvious way to assess how well the Bayes error of the approximate distribution $\mathcal{E}_{\mathrm{Bayes}}(\hat{p})$ estimates the true Bayes error $\mathcal{E}_{\mathrm{Bayes}}(p)$. Theoretical results which bound the distance between these two quantities, perhaps in terms of a divergence $D(p\|\hat{p})$, would be of great interest here.

As detailed in [34], there may be pernicious impacts of the common task framework and the so-called Holy Grail performativity that it induces. For example, a singular focus by the community on the leaderboard performance metrics without regard for any other performance criteria such as fairness or respect for human autonomy. The work here may or may not exacerbate this problem, since trying to approach fundamental Bayes limits is psychologically different than trying to do better than SOTA. As detailed in [33], the shift from competing against others to a pursuit for the fundamental limits of nature may encourage a wider and more diverse group of people to participate in ML research, e.g. those with personality type that has less orientation to competition. It is still to be investigated how to do this, but the ability to generate infinite data of a given target difficulty (yet style of existing datasets) may be used to improve the robustness of classifiers and perhaps decrease spurious correlations.

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
