# Supplementary Material

## A    Proof of Proposition 2

**Proposition 2.** (From main text) *The Bayes error of flow models is monotonically increasing in $\tau$. That is, for $0 < \tau \leq \tau'$, we have that $\mathcal{E}_{Bayes}(\hat{p}_\tau) \leq \mathcal{E}_{Bayes}(\hat{p}_{\tau'})$.*

*Proof.* Note that at temperature $\tau$, the Bayes error is given by

$$\mathcal{E}_{\text{Bayes}}(\hat{p}_\tau) = 1 - \sum_{k=1}^{K} \pi_k \int \prod_{j \neq k} \mathbb{1}(\mathbf{a}_{jk}^\top \mathbf{z} + b_{jk} > 0) \mathcal{N}(d\mathbf{z}; \boldsymbol{\mu}_k, \tau^2 \boldsymbol{\Sigma}) \tag{16}$$

$$= 1 - \sum_{k=1}^{K} \pi_k \int \prod_{j \neq k} \mathbb{1}(\tilde{\mathbf{a}}_{jk}^\top \mathbf{z} + \frac{\tilde{b}_{jk}}{\tau} > 0) \mathcal{N}(d\mathbf{z}; \mathbf{0}, \mathbf{I}) \tag{17}$$

where $\tilde{\mathbf{a}}_{jk} = 2\boldsymbol{\Sigma}^{-1/2}(\boldsymbol{\mu}_k - \boldsymbol{\mu}_j)$, $\tilde{b}_{jk} = (\boldsymbol{\mu}_k - \boldsymbol{\mu}_j)^\top \boldsymbol{\Sigma}^{-1}(\boldsymbol{\mu}_k - \boldsymbol{\mu}_j) \geq 0$. Then it easy to see that for $0 < \tau \leq \tau'$ and $\mathbf{z} \in \mathbb{R}^d$, we have that

$$\prod_{j \neq k} \mathbb{1}(\tilde{\mathbf{a}}_{jk}^\top \mathbf{z} + \frac{\tilde{b}_{jk}}{\tau} > 0) \geq \prod_{j \neq k} \mathbb{1}(\tilde{\mathbf{a}}_{jk}^\top \mathbf{z} + \frac{\tilde{b}_{jk}}{\tau'} > 0) \tag{18}$$

which implies that $\mathcal{E}_{\text{Bayes}}(\hat{p}_\tau) \leq \mathcal{E}_{\text{Bayes}}(\hat{p}_{\tau'})$. $\square$

## B    Further empirical results

### B.1    Hardness of Classes

In addition to measuring the difficulty of classification tasks relative to one another, it also may be of interest to evaluate the relative difficulty of individual classes within a particular task. A natural way to do this is by looking at the error of one-vs-all classification tasks. Specifically, for a given class $j \in \mathcal{K}$, we consider $(\mathbf{x}, 1)$ drawn from the distribution $p_{-j}(\mathbf{x}) = \frac{1}{1-\pi_j} \sum_{i \neq j} \pi_i p_i(\mathbf{x})$, and $(\mathbf{x}, 0)$ from $p_j(\mathbf{x})$. The optimal Bayes classifier in this task is

$$C_{\text{Bayes}}(\mathbf{x}) = \begin{cases} 0 & \text{if } -\log p_j(\mathbf{x}) \leq -\log p_{-j}(\mathbf{x}), \\ 1 & \text{otherwise} \end{cases}.$$

Unfortunately, in this case, the Bayes error cannot be computed with HDR integration, since $p_{-j}$ is now a mixture of Gaussians. However, we can get a reasonable approximation for the error (though less accurate than exact integration would be) in this case using a simple Monte Carlo estimator: $\widehat{\mathcal{E}}_{\text{Bayes}} = \frac{1}{m} \sum_{l=1}^{m} \mathbb{1}(C_{\text{Bayes}}(\mathbf{x}_l) \neq y_l)$, where $y_l \sim \text{Unif}\{0, 1\}$ and $\mathbf{x}_l \mid y_l \sim y_l p_{-j} + (1 - y_l)p_j$ as prescribed above.

The one-vs-all errors by class on CIFAR are shown in Figure 5. It is observed that the errors between the hardest class and the easiest class is huge. On CIFAR-100 the error of the hardest class, squirrel, is almost 5 times that of the easiest class, wardrobe.

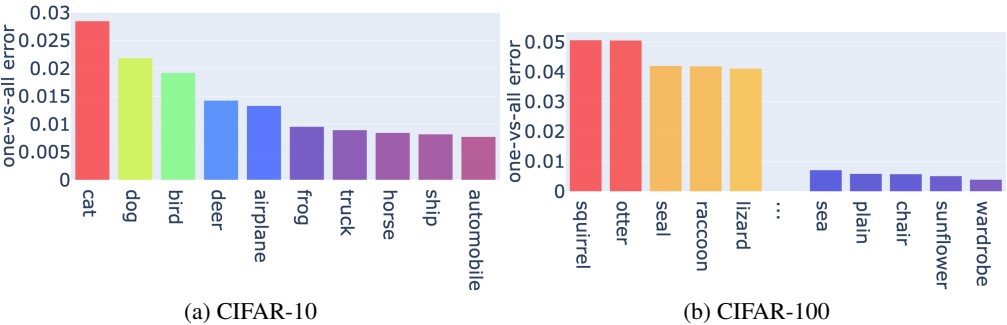

(a) CIFAR-10                    (b) CIFAR-100

Figure 5: Classes Ranked by Hardness

## B.2 Additional samples and Bayes errors from flow models

Below we include examples generating by the trained flow models, and additional datasets generated at different temperatures, and hence Bayes errors.

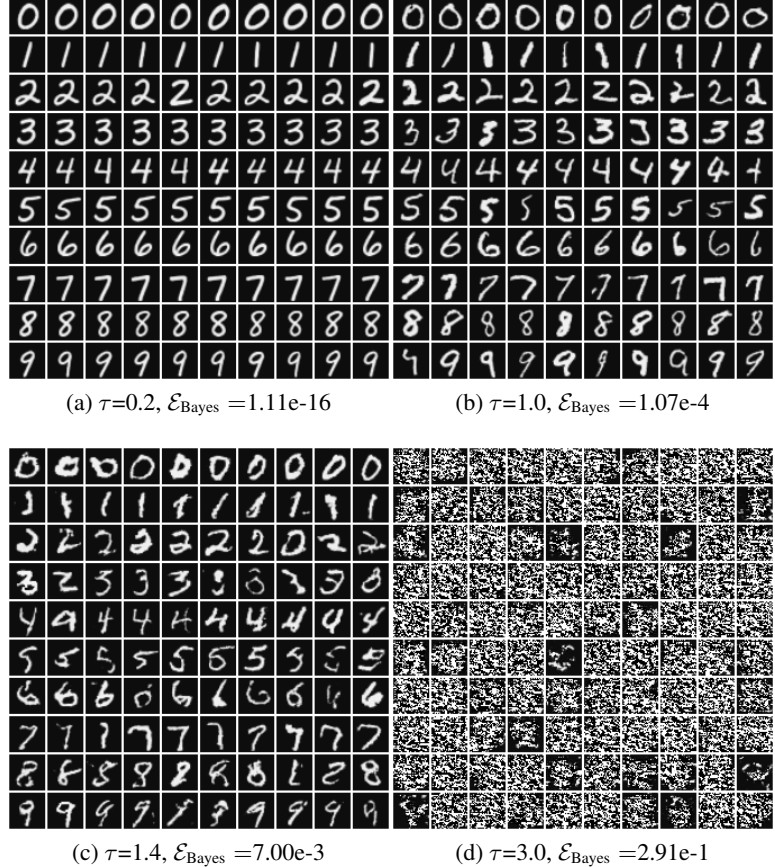

(a) $\tau$=0.2, $\mathcal{E}_{\text{Bayes}}$ =1.11e-16      (b) $\tau$=1.0, $\mathcal{E}_{\text{Bayes}}$ =1.07e-4

(c) $\tau$=1.4, $\mathcal{E}_{\text{Bayes}}$ =7.00e-3      (d) $\tau$=3.0, $\mathcal{E}_{\text{Bayes}}$ =2.91e-1

Figure 6: Generated MNIST Samples with Different Temperatures

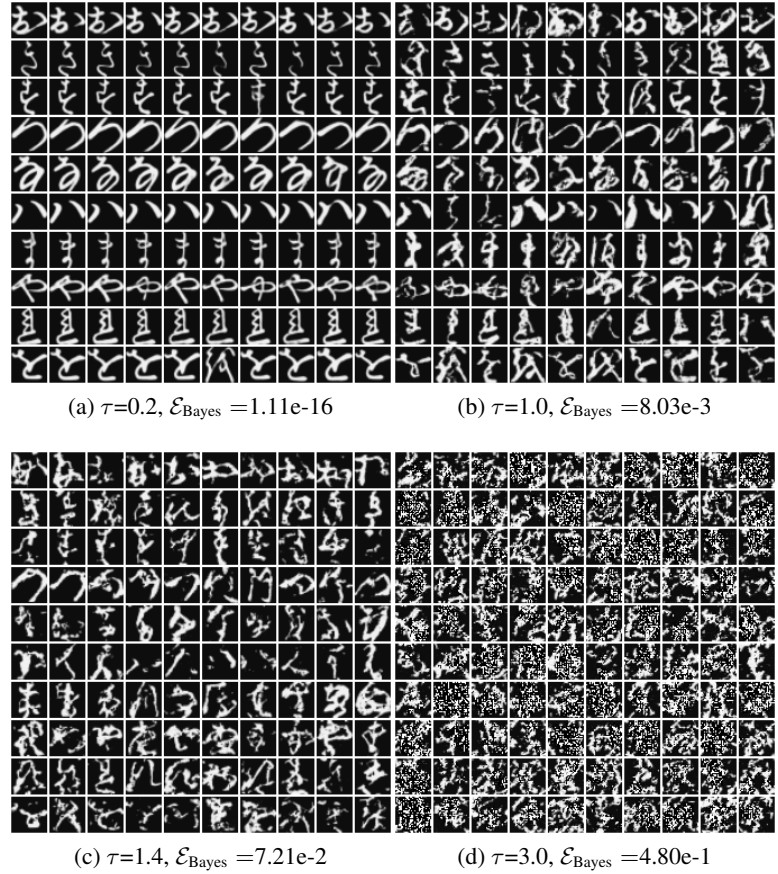

(a) $\tau$=0.2, $\mathcal{E}_{\text{Bayes}}$ =1.11e-16        (b) $\tau$=1.0, $\mathcal{E}_{\text{Bayes}}$ =8.03e-3

(c) $\tau$=1.4, $\mathcal{E}_{\text{Bayes}}$ =7.21e-2        (d) $\tau$=3.0, $\mathcal{E}_{\text{Bayes}}$ =4.80e-1

Figure 7: Generated Kuzushiji-MNIST Samples with Different Temperatures

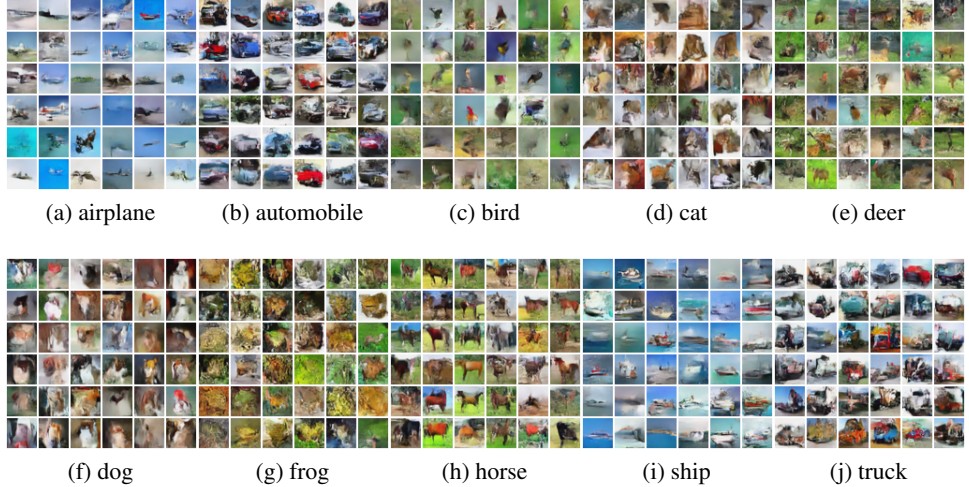

(a) airplane     (b) automobile     (c) bird     (d) cat     (e) deer

(f) dog     (g) frog     (h) horse     (i) ship     (j) truck

Figure 8: Samples generated from conditional GLOW model trained on CIFAR-10.

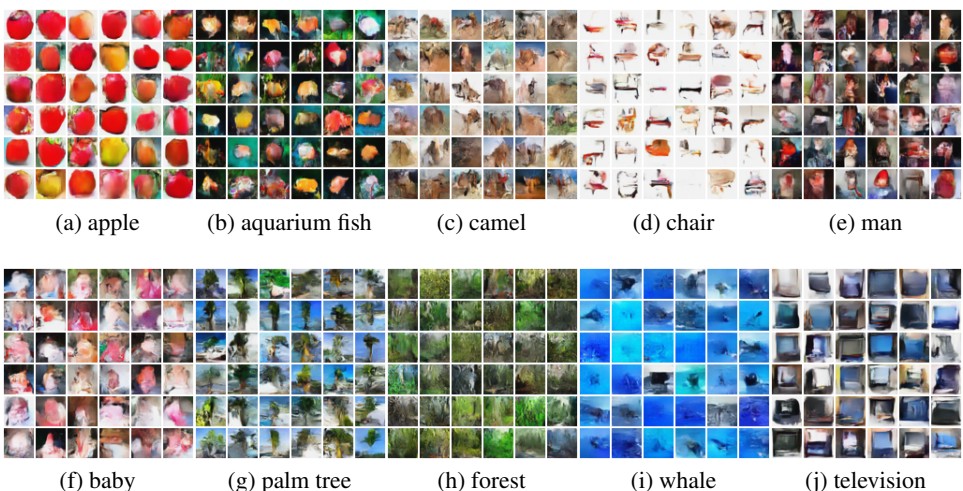

(a) apple    (b) aquarium fish    (c) camel    (d) chair    (e) man

(f) baby    (g) palm tree    (h) forest    (i) whale    (j) television

Figure 9: Samples generated from conditional GLOW model trained on CIFAR-100.

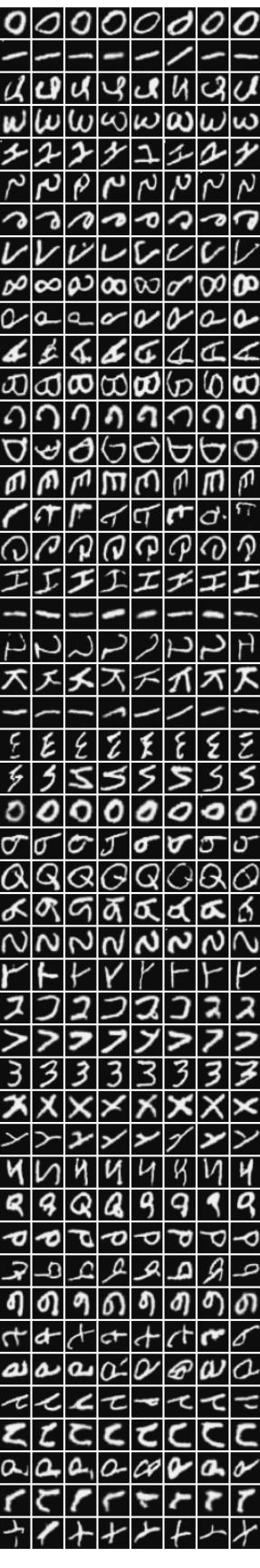

Figure 10: Samples generated from conditional GLOW model trained on EMNIST (balanced). Estimated Bayes Error is 0.09472.