# OpenReview forum: "Evaluating State-of-the-Art Classification Models Against Bayes Optimality"
_NeurIPS.cc/2021/Conference — NeurIPS 2021 Poster_

### Official Review · Reviewer_B5Qe · 2021-07-13

**Rating:** 7
**Confidence:** 5

**Summary:**

This paper proposes using generative models (with known Bayes error) to evaluate classifiers.
By training classifiers on such synthetic distributions, this work is able to compare classifiers against the Bayes-optimal performance, and thus evaluate how close to optimal our current classification methods are.
Further, the paper proposes using similar techniques to evaluate hardness of real distributions.

**Limitations And Societal Impact:**

Yes.

**Main Review:**

## Recommendation
This paper proposes an interesting approach and perspective, and provides adequate experimental support to be useful for future work.
I recommend acceptance, primarily because the idea is interesting, the direction is promising, and the paper is well written.

## Strengths
The question studied in this paper is important: the presence of label noise in real datasets presents a problem when evaluating methods.
Using realistic synthetic datasets, with known Bayes error,
is a reasonable approach to address this issue.

There are many existing works which evaluate generative models using classifiers (e.g. https://arxiv.org/abs/1808.04888)
This work essentially takes the dual approach, and evaluates classifiers using generative models.

Other strengths:
- Paper is well-written, and upfront about limitations.
- Includes several interesting experimental extensions.


## Weaknesses

- Section 4.3, evaluating real dataset-hardness, is somewhat weak in both experimental and theoretical justification.
For example, here is another way to approximate the Bayes error: we simply train the best classifier we can (in practice), and observe its test error.
This approximates the supremum over all functions with a supremum over functions we can find -- it is just another way of approximating the Bayes error. How does this naive approach compare with your approach? Are there advantages of your approach in this case?

- There is no discussion/ablation about how the choice of generative models / GLOW architecture affects the conclusions. For example, the default GLOW architecture is convolutional. Does this give unfair advantage to convolutional classifiers trained on this distribution?
What if the GLOW model is, say, transformer based -- does this change any conclusions?
In general, the proposed method could in theory depend heavily on the choice of generative architecture, so more discussion on this point would be helpful.

- Section 4.2 only reports results for MNIST and F-MNIST. Could more synthetic-dataset results be included? I consider this the most promising direction of this paper, and so more experimental support will strengthen the paper.

- Section 4.2 only evaluates classifiers at a fixed sample size. This seems like a missed opportunity:
with a generative model in hand, one would evaluate the learning curve of classifiers (even for larger data sizes than the original train sets). This could yield further insight into, for example, the scaling laws of classifiers -- or whether certain classifiers
asymptotically approach the Bayes error.

## Citations

This is related to papers which evaluate generative models using classifiers (the "dual approach").
For example, this paper and the references therein: https://arxiv.org/abs/1808.04888

**Time Spent Reviewing:**

1

---

> ### Author Response · Authors · 2021-08-10
> **Response to reviewer 4**
>
> Many thanks to the reviewer for the detailed and useful comments on our work, they will surely help to improve the paper.
>
> With respect to the comments on the results in Section 4.3: while we cannot provide theoretical justification for comparing the Bayes error estimated on synthetic datasets to the SOTA classification errors on the original datasets, we believe that it is nonetheless an interesting comparison. Indeed, the hypothetical alternative approach (training many classifiers and picking the best one) is precisely what the reported SOTA errors (in the 6th column of Table 1) give us. For example, an interesting insight gained from comparing these numbers to the estimated Bayes errors is that the ordering in difficulty between datasets is approximately preserved, which provides a stronger indication of the intrinsic difficulty of these tasks.
>
> We agree with the reviewer that a more comprehensive study of the dependence on the flow architecture would be insightful; we will do our best to provide further empirical studies investigating this. These models tend to be computationally expensive to train, especially given the number of different datasets we consider, but it would likely be feasible to fix a single dataset and perform a study on the effect of different flow architectures.
>
> Indeed, we would like to include studies on more datasets in Section 4.2. The focus on the two MNIST datasets was largely a result of time constraints, but we do plan to include similar results on additional datasets in an updated version of the paper.
>
> We agree completely that investigating the effect of varying training sample size would be a very natural application of our approach. This is a top priority for additional experiments for us, and we absolutely plan to include results on this topic in an updated draft.
>
> Finally, we thank the reviewer for the reference to the “dual approach” to evaluate generative models -- we were not aware of this literature, and will do our best to incorporate a discussion of this topic, and its relation to the present work, as we continue updating the paper.

---

### Official Review · Reviewer_jysn · 2021-07-14

**Rating:** 5
**Confidence:** 3

**Summary:**

The submission makes a very interesting observation: assuming the same per-class prior probabilities in both cases, the Bayes error of class-conditional distributions represented by normalizing flow models is equal to the Bayes error of the corresponding class-conditional base distributions (i.e., it is unchanged by the invertible transformation used in the normalizing flows). In experiments, the submission trains normalizing flow models on standard image classification datasets, using Gaussian base distributions with equal covariance matrices. These normalizing flow models are then applied as sources of realistic synthetic data for which the Bayes error can be approximated. GLOW is used to train the normalizing flow models in the experiments. The submission shows how the difficulty of the synthetic classification problems can be controlled by the temperature parameter of the normalizing flow.  Results for SOTA models on synthetic versions of MNIST and Fashion-MNIST are presented for various levels of the temperature parameter and compared to the Bayes error, demonstrating a gap wrt the Bayes error at large values of the parameter. The submission also shows that ranking the standard image classification problems considered in the submission by "synthetic" Bayes error produces a similar result as ranking by published SOTA performance on the corresponding real-world datasets. Finally, it shows how to rank the difficulty of various classes in a classification problem by considering a 1-vs-rest set-up for the Bayes error.


**Ethical Concerns:**

Not applicable.

**Limitations And Societal Impact:**

Section 4.4 resorts to Monte-Carlo integration. How important is it to use the more efficient integration for the previous results then? Using Monte Carlo integration throughout would mean that the covariance matrices no longer have to be equal. A possibly strong limitation of the results presented in the submission as it stands is that the class-conditional base distributions have equal covariance matrices. (This should also be mentioned in the section detailing the limitations of the work.)


**Main Review:**

The proposed method is a very appealing method for generating synthetic datasets and is based on a very neat observation regarding the Bayes error of distributions transformed using an invertible map. The main question is whether the proposed procedure can be used to evaluate how close the SOTA on various standard problems is to the true Bayes error. This is not the case, as pointed out by the authors themselves. Thus, in my view, the submission needs to be rewritten a bit, making it clearer that the procedure is primarily "just" a method for generating realistic synthetic datasets for which the Bayes error can be computed, and focussing the experiments on comparing state-of-the-art methods on synthetic datasets generated using the method (extending the experiments in Section 4.2 of the current version of the paper). The results regarding dataset difficulty in Table 1 do not seem particularly useful because we do not know how close the Bayes error of the NF model is to the true Bayes error.

Section 4.2 (particularly regarding the results in Figure 4): it would be very useful to know what happens when the amount of training data is increased beyond 60,000.

Table 1: The test NLL is not particularly interpretable. It would be much more useful to compute the classification error of the flow-based models for reference. (The current process used in the submission may have to be changed for this so that the labels in the test data are not used for hyperparameter tuning: Equation 16 requires labels, so the NLL on the test is computed based on labels and the model is chosen by considering the labels.)

What are the sizes of the test sets for the datasets in Table 1?

Table 1: In several cases, the SOTA error is lower than the Bayes error, even on three *MNIST datasets! This reinforces the point made above.

Typos, etc.:

The term "ConvNet" suddenly appears in the text without any previous discussion or a reference.

"a benchmark datasets"

"the distribution ... concentrate"

"still substantial gap"

"To this end"?

"For example, a singular..." -- incomplete sentence



**Time Spent Reviewing:**

1.5 hours

---

> ### Author Response · Authors · 2021-08-10
> **Response to reviewer 3**
>
> We thank the reviewer for their time spent evaluating our work, and for their useful comments on the draft. We agree that the exact objective of the paper can be clarified; we will be sure to do this in an updated draft. We also agree that the experiments in section 4.2 could, and should, be expanded. We will continue to run further experiments on additional datasets and model architectures. With respect to the results listed in Table 1, it is our view that including the SOTA on the original datasets is a useful point of comparison. For example, it is interesting to note that the ordering of difficulty is largely preserved between the SOTA error and the estimated Bayes errors. In particular, for models with low NLL, the Bayes errors align very closely with the SOTA errors, indicating that our approach gives a meaningful measure of the relative hardness ranking across datasets, which could still provide guidance on the dataset design and experimentation. Moreover, as the quality of flow models continues to improve with further research, we suspect that these results will be of even more use. Nonetheless, as we mention in the conclusion, and important step for future work would be to tie a quantity like the NLL more directly to the Bayes error estimation problem.
>
> We also agree with the reviewer that it would be very interesting to evaluate to what extent the test errors approach the Bayes error as we increase the size of the generated training datasets (we also comment on this further in our response to Reviewer 4). We plan to design additional experiments around this question.
>
> We agree the test NLL is not as interpretable as other metrics such as the classification accuracy. On the other hand, to our knowledge, it has been widely used in flow-based model training and the test NNL (which is essentially equivalent to the empirical KL-divergence) is a pretty standard way to evaluate likelihood-based models. For example, in language modeling the most popular metric used in the evaluation is test perplexity, which is a monotonic function of the test NLL.
>
> The size of the test sets generated were chosen to be the same size as the test sets for the original datasets, e.g. 10,000 for the MNIST series datasets.
>
> With respect to the Monte Carlo integration used in section 4.4: this approach is significantly slower and less efficient than the HDR integration routine. Indeed, we attempted this originally, but found that it gave significantly much higher variance results, and required much more time to compute. The results in section 4.4 should be considered in this context. However, we do agree that relaxing the equal covariance assumption would perhaps lead to significant improvements in the quality of the flow models produced. This would require a generalization of the integration technique given in [13], which may indeed be feasible, though we have not been able to experiment with this as of the time of writing.
>
> We also thank the reviewer for pointing out several typos, which we will address in an updated draft of the paper.

---

### Official Review · Reviewer_QpRs · 2021-07-16

**Rating:** 7
**Confidence:** 4

**Summary:**

The goal of this paper is to compute the intrinsic hardness of machine learning benchmark datasets. The approach is based on computing the Bayes error of the underlying data distribution behind the benchmark datasets. There are already many methods that allow estimation of the Bayes error in classification, but most of them compute an approximate Bayes error of the exact distribution, while this paper computes the exact Bayes error of an approximate distribution.

The theoretical contributions of this paper is to first show that the Bayes error can be computed efficiently using a variant of Holmes-Diaconis-Ross integration, but requires an assumption on the data distribution that the class-conditionals are Gaussians. However, the paper proposes to use normalizing flows with Gaussians for the base distributions in order to compute the Bayes error for real-world datasets, with proof showing that the Bayes error is invariant under invertible transformation. Another contribution is to change the ''temperature'' by multiplying a constant to the covariance of the base distribution, to control the Bayes error of flow models.

The experimental contributions of this paper is to use the proposed method for evaluating SOTA models and to check how close it is to the computed Bayes error, and provide a ranking of benchmark datasets based on the Bayes error. The paper also shows a simple way to compute the difficulty of each class within a certain dataset.

**Ethical Concerns:**

I do not see any ethical concerns in this paper.

**Limitations And Societal Impact:**

Yes.

**Main Review:**

### Originality
- The approach of computing the exact Bayes error of an approximate distribution instead of computing an approximate Bayes error of the exact distribution, is original and provides a new direction for Bayes error computation. The application of Holmes-Diaconis-Ross method in order to compute the integrals that appear in the Bayes error w.r.t. Gaussian distributions is interesting, and the idea to use normalizing flows in this setup is also a nice application. It combines existing ideas in a very nice way.

### Quality
- The paper discusses some related works on Bayes error and dataset hardness (mostly in the Introduction section), but offers no comparison with these works in the experiments. Adjusting the temperature seems to be a unique characteristic of the proposed method, but for other experiments, it is not clear why other methods that compute the Bayes error were not considered.
- As stated in the conclusion, a limitation is that there is no way to assess how well the Bayes error of the approximate distribution estimates the true Bayes error.

### Clarity
- The paper is quite clear and well organized.
- I am not sure if I will be able to reproduce the results completely, but it is written that code repo is coming out soon.
- In the introduction, a drawback was discussed (other methods do not work with high-dimensional data), but it seems that the proposed dataset also cannot be used with high-dimensional data due to the memory intensiveness of running the HDR integration routine. Therefore, the story of the paper was somehow vague, since the paper did not solve the original motivation discussed in the introduction.

### Other minor comments
- It is interesting that ConvNets and ResNets start to be much worse than the Bayes error as the temperature becomes higher. I wasn't sure if this can be explained simply by the fact that the Bayes error is higher, and was thinking that it is due to the high distortion of the image as shown in Fig.2(d) and Fig.7(d). Since these architectures are designed for natural images, I was wondering if simple feed-forward NNs will be a better alternative under the high-temperature versions.
- Artifacts are easier and animals are harder in Fig.5. Are there any ideas on why this might be the case?

---

After rebuttal:
Thank you for answering my questions and for the additional comments. I have a better understanding of the contribution of the paper and would like to raise my score. I would like to suggest the authors to include experiments with other Bayes error estimation methods, perhaps with MNIST-sized datasets.

**Time Spent Reviewing:**

4

---

> ### Author Response · Authors · 2021-08-10
> **Response to reviewer 2**
>
> We thank the reviewer for their thoughtful comments on our paper. In what follows, we address some of the main comments/concerns.
>
> Comparison to existing Bayes error estimates: as we mention in the introduction, there have been several proposed estimators for the Bayes error in the literature. The vast majority of these estimators, however, rely on estimation of some type of f-divergence, which is computationally intractable in high dimensions with standard-sized datasets. Nonetheless, there have been a few examples in the literature of the estimators computed on MNIST-sized datasets (e.g. [7]), though certainly none for datasets much larger than that. We agree that including the estimates using these methods would be a useful comparison to include in our work; we will incorporate this into an updated draft of the paper.
>
> Efficiency of the HDR integration routine: the explanation of this issue was not adequately addressed in the text of the current draft, so we thank the reviewer for bringing up this point. To be more specific, the integration method we use (from [13]), is indeed highly efficient relative to existing integration techniques. The issue that arises when the feature space becomes very large (e.g. D = 64x64x3 = 12,288), is the computation and storage of the constraint matrices (a_{jk}) (defined on line 95), which requires as an intermediate step computing a matrix of size D^2 = (12,288)^2. These proved too large to handle in memory on a single machine, and hence we focused on slightly smaller datasets which were feasible to work with. However, the constraint matrices themselves are tractable to store in memory, so we are hopeful that a more efficient approach to computing them may be possible -- we will consider how this might be accomplished as we continue to improve this work.
>
> Feedforward NNs may be better at higher temperatures: this is an interesting idea that we had not considered. Indeed, since state-of-the-art image models are specifically designed to exploit structure in natural images, it may be the case that higher temperature datasets would be better modeled by different architectures. We will investigate this idea in further experiments.
>
> The observation that artifacts tend to be easier than animals is a nice one: we do not have a rigorous explanation yet but it appears like the distributions of animals seem more ``diverse” with a lot of variations/modes, making them harder to approximate and distinguish. This is something that could certainly be investigated more using our technique.

---

### Official Review · Reviewer_tfr3 · 2021-07-17

**Rating:** 7
**Confidence:** 3

**Summary:**

Update: I've read the other reviews and the authors' responses, and I feel this paper is a valuable contribution. I am in favor of accepting and will increase my score to 7.

This paper proposed a new way for benchmarking state-of-the-art models. By leveraging normalizing flows and properties of Bayes error, the authors suggested computing the exact Bayes error of generative models learned using normalizing flows. Empirical investigations on generating synthetic datasets, evaluating SOTA models as well as evaluating data hardness is performed.

**Limitations And Societal Impact:**

Yes.

**Main Review:**

Originality:
I believe this paper brings forth a novel approach to evaluate SOTA models against Bayes optimality. The authors make clear how this approach differs from previous methods where approximation of Bayes error for exact distribution is usually computed. Instead, the authors use normalizing flow to find approximate distributions, where exact Bayes error can be computed. The relative papers seem to be adequately cited. The numerical study of generating synthetic data with desired error level is also very interesting. By comparing the SOTA test errors with exact Bayes errors, the model performance can be evaluated on an absolute scale.

Quality:
The claims appear well supported, with no major technical issues aside from limitations that were discussed. The experiments are also clearly explained. As mentioned in the limitation, it would be interesting to see the performance under other flow models.

Clarity:
This submission is well organized, but some notations need to be clarified. For example $p$ in line 26 and line 38 is used as data distribution, but in Eq.(1),  Eq.(2) and Eq.(3) are used as probability measure.

Line 84: “… are Gaussian: $q_j(x) = \mathcal{N}(z;\mu_j,\Sigma_j)$ …”, do you mean $ q_j(z)$ instead?

Significance:
The topic of evaluating classification models is important. The authors proposed a unique approach to tackle this problem.


**Time Spent Reviewing:**

6

---

> ### Author Response · Authors · 2021-08-10
> **Response to reviewer 1**
>
> We thank the reviewer for positive comments on our work, and helpful feedback. We will address the notational issues pointed out in an updated draft.

---

### Decision · Program_Chairs · 2021-09-28

**Decision:**

Accept (Poster)

**Comment:**

Based on review and discussion,

* The paper is clear,
* the contributions is significant and original,
* the claims are well expressed
* the experiments are well executed.

Hence, I recommend acceptance of this work.


**Consistency Experiment:**

NeurIPS has a long history of experimentation. In 2014, NeurIPS ran an experiment in which 10% of submissions were reviewed by two independent committees to quantify the randomness in the review process. This year, we repeated a variant of this experiment to see how the quality of the review process has changed over time.  This paper was part of the experiment and was therefore assigned to two committees (consisting of reviewers, an Area Chair, and a Senior Area Chair) that reached independent decisions.  If both committees made the same recommendation, this recommendation was followed. If a single committee recommended acceptance, the paper was accepted (with the exception of a few cases in which the other committee identified what we considered a fatal flaw, e.g., an error in a key result).

This copy’s committee reached the following decision: **Accept (Poster)**

The other committee assigned to the paper recommended **Reject**.  You can find the other set of reviews, along with any follow up discussion with the authors here:
https://openreview.net/forum?id=Xa3-BmKfN0R